# Exploring Alternative Pathways to Target Bacterial Type II Topoisomerases Using NBTI Antibacterials: Beyond Halogen-Bonding Interactions

**DOI:** 10.3390/antibiotics12050930

**Published:** 2023-05-18

**Authors:** Maja Kokot, Doroteja Novak, Irena Zdovc, Marko Anderluh, Martina Hrast, Nikola Minovski

**Affiliations:** 1Laboratory for Cheminformatics, Theory Department, National Institute of Chemistry, Hajdrihova 19, 1000 Ljubljana, Slovenia; maja.kokot@ki.si; 2Department of Pharmaceutical Chemistry, Faculty of Pharmacy, University of Ljubljana, Aškerčeva cesta 7, 1000 Ljubljana, Slovenia; doroteja.novak@ffa.uni-lj.si (D.N.); marko.anderluh@ffa.uni-lj.si (M.A.); 3Institute of Microbiology and Parasitology, Veterinary Faculty, University of Ljubljana, Gerbičeva 60, 1000 Ljubljana, Slovenia; irena.zdovc@vf.uni-lj.si

**Keywords:** DNA gyrase, topoisomerase IV, antibacterial, NBTIs, halogen-bonding interactions, hydrogen-bonding interactions, van der Waals interactions

## Abstract

Novel bacterial topoisomerase inhibitors (NBTIs) are a new class of antibacterial agents that target bacterial type II topoisomerases (DNA gyrase and topoisomerase IV). Our recently disclosed crystal structure of an NBTI ligand in complex with DNA gyrase and DNA revealed that the halogen atom in the para position of the phenyl right hand side (RHS) moiety is able to establish strong symmetrical bifurcated halogen bonds with the enzyme; these are responsible for the excellent enzyme inhibitory potency and antibacterial activity of these NBTIs. To further assess the possibility of any alternative interactions (e.g., hydrogen-bonding and/or hydrophobic interactions), we introduced various non-halogen groups at the *p*-position of the phenyl RHS moiety. Considering the hydrophobic nature of amino acid residues delineating the NBTI’s binding pocket in bacterial topoisomerases, we demonstrated that designed NBTIs cannot establish any hydrogen-bonding interactions with the enzyme; hydrophobic interactions are feasible in all respects, while halogen-bonding interactions are apparently the most preferred.

## 1. Introduction

Bacterial type II topoisomerases (DNA gyrase and topoisomerase IV) are well-validated targets for antibacterial chemotherapy. They are vital bacterial enzymes involved in the maintenance of the correct spatial topological state(s) of DNA molecule, which is of fundamental importance for the major biological processes in bacteria such as DNA replication, transcription, and control of gene expression [1]. Despite the high level of structural resemblance that DNA gyrase and topoisomerase IV (topoIV) share, they are substantially involved in distinct intracellular functions in bacteria; DNA gyrase is responsible for the introduction of negative supercoils into the DNA molecule, while topoIV is responsible for DNA’s decatenation activity [2]. Consequently, inhibition of the function of either or both enzymes leads to distortion in the correct spatial DNA topology, resulting in death of the bacterial cell [2,3]. These heterotetrameric enzymes are composed of two GyrA and two GyrB subunits (A_2_B_2_ in DNA gyrase), i.e., two ParC and two ParE subunits (C_2_E_2_ in topoIV) [2,4]. A widely known class of antimicrobial agents targeting these bacterial enzymes are fluoroquinolones [5,6]. Some representatives of this class of antibacterials, including ciprofloxacin, levofloxacin, and moxifloxacin, are still successfully used in the clinical practice, but unfortunately their extensive and/or inappropriate application has led to acquired resistance in bacteria [5,7,8]. Therefore, there is an urgent need for development of new antibacterial alternatives for combating bacterial resistance.

Nearly two decades ago, a new class of antibacterial agents called “Novel Bacterial Topoisomerase Inhibitors” (NBTIs) was discovered [9,10,11] (representatives on Figure 1). Similarly to fluoroquinolones, they inhibit the same targets in bacteria by binding to an alternative, non-overlapping binding site in DNA gyrase/topoIV, thereby avoiding cross-resistance to fluoroquinolones [11,12]. In addition to structural differences, NBTIs also have a different mechanism of DNA gyrase/topoIV inhibition. As depicted in Figure 2a, one can perceive that NBTIs are composed of three key parts: an upper left hand side (LHS) that intercalates between the central DNA base pairs of the bacterial DNA, the lower right hand side (RHS) that binds into a deep, non-catalytic hydrophobic binding pocket formed at the interface of both GyrA subunits of DNA gyrase or both ParC subunits of topoIV enzyme, and a central linker moiety connecting both parts [13]. Figure 1 shows representatives of NBTIs with variety of LHS, linker, and RHS moieties [10,12,14,15,16,17]. Viquidacin (NXL-101, **i**) was the first NBTI that underwent phase I clinical trials; however, it was discontinued due to its cardiotoxicity, which manifested as prolonged QT intervals [18]. The most advanced NBTI is gepotidacin, which is currently finishing the third phase of clinical trials for the treatment of uncomplicated urogenital gonorrhea [19] and uncomplicated urinary tract infections commonly caused by *Escherichia coli* [20].

Our recently disclosed crystal structure of *Staphylococcus aureus* DNA gyrase in complex with a potent NBTI ligand (AMK-12, IC_50_ = 0.035 nM) revealed that the halogen atom at the *para* position of the phenyl RHS moiety is able to establish strong symmetrical bifurcated halogen bonds with the backbone carbonyl oxygens of Ala68 residues of both GyrA subunits (PDB ID: 6Z1A, Figure 2b) [21]. To evaluate and re-confirm the significance of the halogen bonds that *p*-halogenated phenyl RHSs form with the enzymes, making them accountable for the excellent enzyme inhibitory potencies of NBTIs against DNA gyrase and topoIV [22], we opted to further probe the possibility and potential impact of some alternative interactions (e.g., hydrogen-bonding and/or hydrophobic interactions) on the NBTIs’ inhibitory potencies and antibacterial activities. For this purpose, a small, targeted library of new NBTI analogues was designed by substituting the halogen atom at the para position of NBTIs RHSs with non-halogen groups (e.g., *p*-methyl and *p*-amino fragments, Figure 1).

## 2. Results and Discussion

With the intention of investigating whether the newly designed NBTI analogues comprising non-halogenated RHS moieties (Table 1) are able to establish any hydrogen-bonding or hydrophobic interaction with the amino acid residues delineating an NBTI’s binding site, a series of flexible molecular docking calculation trials were performed within the NBTI binding site of *S. aureus* DNA gyrase enzyme (PDB ID: 6Z1A) [17]. The molecular docking calculations of NBTI analogues comprising a *p*-amino phenyl RHS moiety (e.g., Table 1, **9**–**11**) revealed that the amino group could not establish any reasonable hydrogen-bonding interactions with the backbone carbonyl oxygens of Ala68 residues, primarily due to its specific spatial geometry. This is clearly demonstrated in Figure 3a, depicting the hydrogen-bonding distances and angles obtained post-docking (*d*_1_ = 2.75 Å and *θ*_2_ = 107°, i.e., *d*_2_ = 3.82 Å and *θ*_2_ = 118°), whose results are simply not in line with the crystallographic evidence of these types of hydrogen bonds in biological systems (e.g., *d* < 2.72 Å and *θ* = 150°–110°) [23]. In contrast to this, NBTI analogues comprising a methyl group at the para position on the phenyl RHS moiety (e.g., Table 1, **5**–**8**) are able to establish some hydrophobic interactions with amino acid residues delineating the NBTI’s binding site (e.g., Ala68, Val71, Met75, and Met121 in *S. aureus* DNA gyrase), which are mainly hydrophobic in nature (Figure 3b).

To check the validity of the molecular docking predictions, our focused library of newly designed NBTIs was synthetized and subsequently biologically evaluated in vitro. All details related to the various synthetic procedures employed for obtaining LHS-linker constructs are thoroughly described in our previous publication [24], while the final compounds (Table 1) were synthesized utilizing a general reductive amination procedure described in detail in the experimental section. 

The in vitro evaluated inhibitory potencies of the synthesized NBTIs on bacterial type II topoisomerase enzymes (DNA gyrase and topoIV) originating from *S. aureus* and *E. coli*, respectively, and their selectivity against human topoisomerase IIα enzyme are listed in Table 1, while their antibacterial activities are summarized in Table 2. 

As shown in Table 1, one can observe that the experimentally determined in vitro on-target potencies of this series of NBTI analogues are in line with the predictions derived from molecular docking calculations. Both the in vitro enzyme inhibitory (Table 1) and antibacterial activity data (Table 2) of NBTI analogues comprising a *p*-amino phenyl RHS moiety (**9**–**11**) clearly demonstrate that these compounds completely abolished the enzyme inhibitory (IC_50_ > 100 µM) and antibacterial activity in both Gram-positive and Gram-negative bacteria (MIC > 128 µg/mL). This clearly pinpoints that *p*-amino phenyl NBTI analogues (**9**–**11**) are not able to establish any hydrogen-bonding interaction with GyrA A68 residues due to the inappropriate spatial conformation of the *p*-amino group (i.e., unsuitable bond angles and geometry), as previously predicted by the molecular docking (Figure 3a). Moreover, considering that the amino group itself is hydrophilic, and the NBTI’s binding pocket in bacterial type II topoisomerases is strictly hydrophobic, there is also a low probability even of a hydrophobic interaction due to the desolvation penalties that negatively affect hydrophobic interactions. Contrary to the NBTI analogues containing *p*-amino phenyl RHS fragments (**9**–**11**), NBTIs comprising *p*-methyl phenyl RHSs (**5**–**8**) establish some hydrophobic contact with the hydrophobic amino acid residues delineating the NBTI’s binding site in bacterial type II topoisomerases (e.g., Ala68, Val71, Met75, and Met121 in *S. aureus* DNA gyrase), as depicted in Figure 3b. These compounds show strong antibacterial activity against Gram-positive bacteria, and potent enzyme inhibition of *S. aureus* DNA gyrase, (Table 1 and Table 2), while in Gram-negative bacteria, they exhibit weaker enzyme inhibitory and antibacterial activity. This implies that hydrophobic interactions apparently play an important role in DNA gyrase inhibition in Gram-positive bacteria. Furthermore, comparison of compounds containing *p*-methyl phenyl RHSs (**5**–**8**) and *p*-bromo phenyl RHSs (**1**–**4**) clearly indicates that bromo analogues exhibit stronger enzyme inhibitory (1- to 12-fold) and antibacterial activity (1- to 64-fold), a result that unequivocally implies the pivotal role and importance of halogen bonds for the excellent antibacterial potencies, as previously confirmed [17,18,21]. It can also be noted that topoIV is probably the primary target in *E. coli*, as it is observed that compounds with IC_50_s < 100 nM on *E. coli* topoIV exhibit strong antibacterial activities (lower MICs) against *E. coli* (Table 1 and Table 2) as well. Moreover, it seems that hydrophobic interactions are also favourable for inhibiting topoisomerase enzyme(s) in the hard-to-treat Gram-negative bacillus *Acinetobacter baumannii,* as particularly exemplified by the stronger antibacterial activities of compounds **5** and **8,** relative to those determined for our previously published compounds (**1**–**4**). Although compounds **5**, **6** and **7** show similar inhibition of *E. coli* DNA gyrase, only compound **5** has potent antibacterial activity against *E. coli*. Compounds with cyclohexane (**6**) and tetrahydropyran (**7**) linkers may have lower permeability and are more easily pumped out by the efflux pumps. Compared with wild-type *E. coli*, the activity was enhanced in the *AcrA* knockout *E. coli* N43 strain and in *E. coli* D22 strain, with a mutation in the lpxC gene that increases membrane permeability. This suggests that the membrane permeability of the compounds is suboptimal for crossing the membrane of Gram-negative pathogens and, more importantly, that they are efficiently pumped out by the *E. coli* efflux pumps. Moreover, the comparison of compounds **1** and **12** shows that the CF_3_ group on the *meta*-position of *p*-bromo phenyl RHS leads to stronger inhibition of topoIV, but a weaker inhibition of DNA gyrase. As expected, gepotidacin shows stronger inhibition of both enzymes (DNA gyrase and topoIV) in Gram-negative bacteria such as *E. coli*, relative to the newly synthetized NBTIs, but also a weaker inhibition in Gram-positive bacteria such as *S. aureus*. According to cytotoxicity assessments on human umbilical vein endothelial cells (HUVECs) and HepG2 liver cancer cell lines (Table 3), compounds **5**, **8** and **12** showed weaker effects by one to three orders of magnitude on human cells compared to antibacterial activity against Gram-positive bacteria (Appendix A), and the same order of magnitude compared to Gram-negative bacteria (Appendix A). Due to the weak antibacterial activity of other compounds (**6**, **7**, **9**–**11**) against Gram-positive and Gram-negative bacteria, MICs were approximately of the same order of magnitude as their cytotoxic IC_50_ values.

Since NBTIs stop the enzymatic reaction at the level of single-strand DNA breaks, there is a correlation between IC_50_ supercoiling and single-strand breaks. They disrupt the spatial symmetry and topology of nascent DNA due to the asymmetry of the 6-methoxy-1,5-naphthyridine LHS intercalated between DNA base pairs. Consequently, they stabilize single-strand DNA breaks in DNA gyrase in a similar manner to the majority of NBTIs, as we confirmed for structurally similar NBTIs in a DNA gyrase cleavage assay in our previous publication [21].

**Table 1 antibiotics-12-00930-t001:** *S. aureus* and *E. coli* DNA gyrase/topoIV inhibitory potencies and human topoisomerase IIα residual activities.

Cmpd	Structure	IC_50_ (µM) ^1^				Human TopoIIα (% RA at 100 µM) ^4^
		DNA Gyrase ^2^		topoIV ^3^		
		*S. aureus*	*E. coli*	*S. aureus*	*E. coli*	
**1 ***	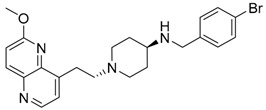	0.026 ± 0.010	0.570 ± 0.06	7.22 ± 0.55	0.042 ± 0.003	98 ± 3
**2 ****	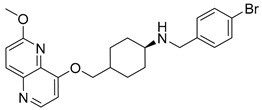	0.372 ± 0.002	4.805 ± 0.774	>100	0.495 ± 0.184	106 ± 3
**3 ****	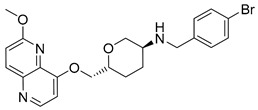	0.294 ± 0.100	7.742 ± 2.308	>100	0.131 ± 0.040	105 ± 1
**4 ****	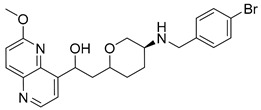	0.065 ± 0.002	1.575 ± 0.404	>100	0.052 ± 0.003	106 ± 0
**5**	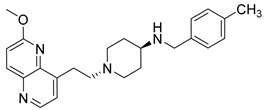	0.145 ± 0.014	1.019 ± 0.005	1.226 ± 0.258	0.089 ± 0.019	103 ± 0
**6**	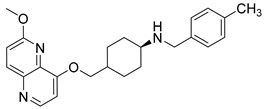	0.403 ± 0.020	1.519 ± 0.361	>100	2.886 ± 0.880	101 ± 0
**7**	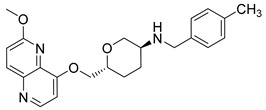	0.194 ± 0.018	1.309 ± 0.348	>100	1.527 ± 0.043	101 ± 0
**8**	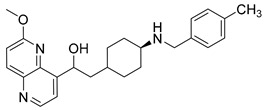	0.077 ± 0.010	1.315 ± 0.460	2.502 ± 0.252	0.083 ± 0.000	100 ± 1
**9**	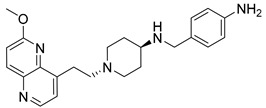	1.383 ± 0.108	>100	>100	5.698 ± 0.014	100 ± 1
**10**	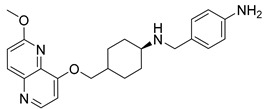	>100	>100	>100	>100	99 ± 1
**11**	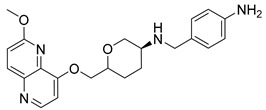	>100	>100	>100	>100	100 ± 0
**12**	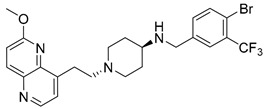	0.062 ± 0.000	0.084 ± 0.020	0.345 ± 0.025	0.023 ± 0.006	102 ± 0
**Gepo**	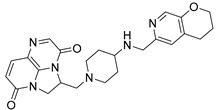	0.374 ± 0.019	0.244 ± 0.040	8.299 ± 0.361	0.049 ± 0.003	ND

^1^ IC_50_, mean of two independent measurements ± SD; ^2^ DNA gyrase supercoiling inhibition assay; ^3^ topoIV relaxation inhibition assay; ^4^ Residual activity, mean ± SD of residual activity (%) at 100 µM compound from two independent experiments ND: not determined; Gepo: gepotidacin. * [25]; ** [24].

**Table 2 antibiotics-12-00930-t002:** Antimicrobial activity of newly designed NBTIs against a panel of Gram-positive and Gram-negative bacterial pathogens.

Cmpd	MIC (µg/mL)												
	1 *	2 **	3 **	4 **	5	6	7	8	9	10	11	12	Gepo
*S. aureus* (ATCC 29213)	0.125	0.125	0.125	0.016	0.032	1	0.5	0.063	64	>128	>128	0.032	0.125
MRSA (QA-11.7) ^1^	0.5	0.25	0.5	0.016	0.016	1	1	0.125	ND	ND	ND	0.063	0.063
MRSA(QA-12.1) ^2^	ND	0.25	0.125	0.016	0.016	2	0.5	0.063	ND	ND	ND	0.032	0.125
MRSA (QA-11.2)	ND	0.5	0.25	0.031	0.032	0.5	0.25	0.032	ND	ND	ND	0.063	0.5
*E. coli* (ATCC 25922)	2	64	128	16	2	>128	>128	8	>128	>128	>128	4	1
*E. coli* D22 ^3^	0.125	8	8	1	1	32	32	1	>128	>128	>128	0.25	0.125
*E. coli* N43 ^4^ (CGSC# 5583)	0.125	1	8	0.25	0.125	4	4	1	64	>128	>128	0.063	0.016
*E. coli* ESBL QA-11.3	ND	128	>128	>128	4	>128	>128	8	>128	>128	>128	4	ND
*K. pneumoniae*	ND	128	128	128	16	>128	>128	16	>128	>128	>128	16	8
*P. aeruginosa* RDK 184	128	32	128	128	>128	>128	>128	128	>128	>128	>128	>128	8
*E. faecalis* RDK 057	0.5	1	2	0.5	0.125	8	4	0.25	>128	>128	>128	0.5	2
VRE	2	1	1	0.032	0.032	2	2	0.125	ND	ND	ND	0.25	1
*A. baumannii*	ND	128	>128	>128	1	64	64	2	>128	>128	>128	2	8

^1^ Resistant to cefoxitin, ciprofloxacin, clindamycin, erythromycin, tetracycline, thiamulin, trimethoprim. ^2^ Resistant to cefoxitin, gentamicin, kanamycin, rifampicin, streptomycin, sulfamethoxazole, tetracycline. ^3^ With a mutation in the *lpxC* gene that increases membrane permeability. ^4^ With *AcrA* knockout (cell membrane efflux pump). * [25]; ** [24]. ATCC, American Type Culture Collection; CGSC, Coli Genetic Stock Centre; QA, Quality Assurance; Gepo: gepotidacin.

**Table 3 antibiotics-12-00930-t003:** Cytotoxicity data for human HUVEC and HepG2 cell lines.

Cmpd	IC_50_ [μM]	
	HUVEC	HepG2
**5**	36.54 ± 9.82	19.99 ± 3.01
**6**	36.65 ± 7.91	22.46 ± 2.70
**7**	>50	>50
**8**	33.07 ± 6.13	22.28 ± 0.36
**9**	34.04 ± 16.87	26.14 ± 9.52
**10**	>50	>50
**11**	>50	>50
**12**	ND	12.00 ± 1.86

IC_50_, means of three independent measurements ± SD. ND = not determined; IC_50_ could not be determined due to a steep change in metabolic activity (MA) from 25.0 μM (0.68 ± 1.30 % MA) to 12.5 μM (99.63 ± 7.48 % MA).

The SWISSADME tool [26] was used to calculate the physicochemical properties of the newly synthesized NBTIs (Table 4). With the exception of compound **12**, they exhibit similar physicochemical properties. Despite the higher molecular weight, larger number of heavy atoms, rotatable bonds, and H-bond acceptors, it shows strong biological activity. The major difference between compounds with *p*-methyl phenyl RHSs (**5**–**8**) and *p*-amino phenyl RHSs (**9**–**11**) is in their topological polar surface area (TPSA) parameter, suggesting that a larger polar surface area leads to lower biological activity.

**Table 4 antibiotics-12-00930-t004:** Physicochemical properties of the newly synthesized NBTIs.

Cmpd	5	6	7	8	9	10	11	12
Formula	C_24_H_30_N_4_O	C_24_H_29_N_3_O_2_	C_23_H_29_N_5_O	C_25_H_31_N_3_O_2_	C_23_H_29_N_5_O	C_23_H_28_N_4_O_2_	C_22_H_26_N_4_O_3_	C_24_H_26_BrF_3_N_4_O
Molecular weight [g/mol]	390.52	391.51	393.48	405.53	391.51	392.49	394.47	523.39
Num. heavy atoms	29	29	29	30	29	29	29	33
Num. arom. heavy atoms	16	16	16	16	16	16	16	16
Fraction Csp3	0.42	0.42	0.39	0.44	0.39	0.39	0.36	0.42
Num. rotatable bonds	7	7	7	7	7	7	7	8
Num. H bonds acceptors	5	5	6	5	5	5	6	8
Num. H bonds donors	1	1	1	2	2	2	2	1
Molecular Refractivity	121.60	116.31	112.59	120.75	121.04	115.75	112.03	129.33
TPSA [Å²]	50.28	56. 27	65.50	67.27	76.30	82.29	91.52	50.28

The physicochemical properties were calculated with the SWISSADME tool [26].

Different RHS moieties form different interactions with the amino acid residues of the binding pocket, resulting in a different structure–activity relationship (SAR). As shown in Figure 4, NBTIs with *p*-bromo phenyl RHSs show inhibition for three out of four enzymes, due to hydrophobic interactions and additional strong halogen bonds in the binding pocket. Compounds with *p*-methyl phenyl RHSs that form a hydrophobic interaction show inhibition, but it is most likely weaker, as can be deduced from the enzyme inhibition data. Since no inhibition was observed for *p*-amino phenyl RHS NBTIs, we presume that they fail to form expected hydrogen bonds with backbone carbonyl oxygens of Ala68 in the enzyme-binding pocket.

## 3. Materials and Methods

### 3.1. Molecular Docking Calculations

Molecular docking calculations of newly designed NBTIs within the binding site of *S. aureus* DNA gyrase were performed by employing the GOLD docking suite in a flexible manner [27]. Our recently disclosed crystal structure of *S. aureus* DNA gyrase in complex with a potent NBTI ligand (AMK-12; PDB ID: 6Z1A) [21] was used for the calculations. The experimental coordinates of the co-crystallized ligand (AMK-12) were used for defining the NBTI’s binding site–cavity radius of 15.5 Å for *S. aureus* DNA gyrase. Each of the NBTI compounds was docked into the defined binding site up to 10 times using the GOLD genetic algorithm (the settings and parameters of which are shown in Appendix A). During the calculations, the amino acid residues, Met75, Asp83, and Met121, in *S. aureus* GyrA were flexible. The protocol for molecular docking was verified by conducting three re-docking validation trials of the co-crystallized AMK-12 ligand within the defined binding site (Appendix A and Appendix A) [28]. As a criterion for well-tuned structure-based settings, the heavy atoms’ root mean square deviation values (RMSD ≤ 2.0 Å) between each calculated AMK-12 docking solution and the co-crystallized AMK-12 conformation were calculated (Appendix A) [28]. To check the correct spatial orientation and conformation of the docked NBTIs in the binding site relative to co-crystallized AMK-12 ligand conformation, they were also visually examined. 

### 3.2. General Chemical Methods

The general chemical methods (chemicals, thin layer chromatography (TLC) analysis, NMR spectroscopy, high resolution mass spectra HPLC analysis) were the same as in our previous publication [25].

### 3.3. Synthesis

#### 3.3.1. Synthesis of Intermediates

Intermediates 1-(2-(6-Methoxy-1,5-naphthyridin-4-yl)ethyl)piperidin-4-amine (Figure 5A), (1R,4R)-4-(((6-methoxy-1,5-naphthyridin-4-yl)oxy)methyl)cyclohexan-1-amine (Figure 5B), (3R,6S)-6-(((6-methoxy-1,5-naphthyridin-4-yl)oxy)methyl)tetrahydro-2H-pyran-3-amine (Figure 5C), and 2-((1R,4R)-4-aminocyclohexyl)-1-(6-methoxy-1,5-naphthyridin-4-yl)ethan-1-ol (Figure 5D) (Figure 5) were synthetized according to the procedure in our previous publication [24].

#### 3.3.2. General Procedure for Reductive Amination

Acetic acid (2 drops) was added to a solution of Figure 5A–D (1 eq) and suitable *p*-substituted benzaldehyde (1.3 eq) dissolved in methanol (~5 mL). The mixture was stirred for 1 h at room temperature. Then, Na(CH_3_COO)_3_BH (4 eq) was added to the reaction mixture and was stirred for 16 h at room temperature. The solvent was evaporated and the crude product was extracted with in ethyl acetate (20 mL) and 3 × 10 mL of 0.1 M NaOH. The organic phase was dried over Na_2_SO_4_, filtered and evaporated. The product was purified by flash chromatography on silica gel to afford the title compounds (**5**–**12**).

#### 3.3.3. Synthesis of the Final Compounds

##### 1-(2-(6-Methoxy-1,5-naphthyridin-4-yl)ethyl)-*N*-(4-methylbenzyl)piperidin-4-amine (**5**)

The title compound was obtained according to general procedure using **A** (91 mg, 0.318 mmol, 1 eq), 4-methylbenzaldehyde (46 mg, 0.38 mmol, 1.2 eq), acetic acid, DCM (~5 mL) and Na(CH_3_COO)_3_BH (101 mg, 0.477 mmol, 1.5 eq). The crude product was purified using flash column chromatography (SiO_2_, dichlorometane: methanol = 20: 1 to 9:1) to afford **5** (51 mg, 41%). ^1^H NMR (400 MHz, CDCl_3_) *δ* 8.63 (d, *J* = 4.5 Hz, 1H), 8.16 (d, *J* = 9.0 Hz, 1H), 7.38 (d, *J* = 4.5 Hz, 1H), 7.22 (d, *J* = 7.9 Hz, 2H), 7.12 (d, *J* = 7.8 Hz, 2H), 7.08 (d, *J* = 9.0 Hz, 1H), 4.05 (s, 3H), 3.78 (s, 2H), 3.35 (dd, *J* = 9.4, 6.6 Hz, 2H), 3.04 (d, *J* = 11.7 Hz, 2H), 2.79–2.74 (m, 2H), 2.63–2.51 (m, 1H), 2.31 (s, 3H), 2.16 (t, *J* = 10.7 Hz, 2H), 1.96 (d, *J* = 11.7 Hz, 2H), 1.53 (td, *J* = 13.9, 3.5 Hz, 2H) ppm. “Amine H is exchangeable and is not visible on NMR spectrum.” ^13^C NMR (100 MHz, CDCl_3_): *δ* 161.41, 147.67, 146.61, 141.46, 140.95, 140.30, 136.71, 136.64, 129.16, 128.22, 124.23, 116.27, 58.32, 53.82, 53.70, 52.17, 50.18, 32.22, 28.39, 21.10 ppm. HRMS: *m*/*z*: calcd for C_24_H_31_0N_4_O [M + H]^+^: 391.2492, found: 391.2484. HPLC: t_R_ = 4.537 min (97 % at 254 nm).

##### (4-(((6-Methoxy-1,5-naphthyridin-4-yl)oxy)methyl)-*N*-(4-methylbenzyl)cyclohexan-1-amine (**6**)

The title compound was obtained according to general procedure using **B** (105 mg, 0.365 mmol, 1 eq), 4-methylbenzaldehyde (52 mL, 0.438 mmol, 1.2 eq), acetic acid, DCM (~5 mL) and Na(CH_3_COO)_3_BH (116 mg, 0.548 mmol, 1.5 eq). The crude product was purified using flash column chromatography (SiO_2_, dichlorometane: methanol = 20: 1 to 9:1) to afford **6** (86 mg, 60%). ^1^H NMR (400 MHz, CDCl_3_): *δ* 8.58 (d, *J* = 5.2 Hz, 1H), 8.13 (d, *J* = 9.0 Hz, 1H), 7.22 (d, *J* = 7.9 Hz, 2H), 7.12 (d, *J* = 7.8 Hz, 2H), 7.09 (d, *J* = 9.0, Hz, 1H), 6.85 (d, J = 5.3, 1H), 4.08 (s, 3H), 3.99 (d, *J* = 6.1 Hz, 2H), 3.80 (s, 2H), 2.57–2.50 (m, 1H), 2.32 (s, 3H), 2.11–2.00 (m, 5H), 1.34–1.15 (m, 4H) ppm. ^13^C NMR (100 MHz, CDCl_3_): *δ* 161.47, 160.01, 148.82, 142.61, 139.97, 136.67, 134.38, 129.17, 128.25, 116.48, 104.96, 73.70, 56.19, 53.62, 50.56, 37.18, 32.39, 28.30, 21.11 ppm. HRMS: *m*/*z*: calcd for C_24_H_30_N_3_O_2_ [M+H]^+^: 391.2333, found: 391.2325. HPLC: t_R_ = 5.717 min (100 % at 254 nm).

##### 6-(((6-Methoxy-1,5-naphthyridin-4-yl)oxy)methyl)-*N*-(4-methylbenzyl)tetrahydro-2*H*-pyran-3-amine (**7**)

The title compound was obtained according to general procedure using **C** (124 mg, 0.429 mmol, 1 eq), 4-methylbenzaldehyde (61 mL, 0.514 mmol, 1.2 eq), acetic acid, DCM (~5 mL) and Na(CH_3_COO)_3_BH (136 mg, 0.643 mmol, 1.5 eq). The crude product was purified using flash column chromatography (SiO_2_, dichlorometane: methanol = 20: 1 to 9:1) to afford **7** (113 mg, 67%). ^1^H NMR (400 MHz, CDCl_3_): *δ* 8.58 (d, *J* = 5.2 Hz, 1H), 8.13 (d, *J* = 9.0 Hz, 1H), 7.19 (d, *J* = 8.0 Hz, 2H), 7.12 (d, *J* = 7.9 Hz, 2H), 7.09 (d, *J* = 9.0 Hz, 1H), 6.90 (d, *J* = 5.3 Hz, 1H), 4.29–4.25 (m, 1H), 4.17–4.10 (m, 2H), 4.08 (s, 3H), 3.88–3.81 (m, 1H), 3.81–3.74 (m, 2H), 3.17 (t, *J* = 10.6 Hz, 1H), 2.78–2.70 (m, 1H), 2.32 (s, 3H), 2.18–2.13 (m, 1H), 1.98–1.92 (m, 1H), 1.68–1.56 (m, 1H), 1.43–1.31 (m, 1H) ppm. ^13^C NMR (100 MHz, CDCl_3_): *δ* 161.53, 159.81, 148.83, 142.76, 140.04, 137.43, 136.69, 134.38, 129.21, 128.02, 116.61, 105.50, 75.52, 72.74, 71.73, 53.71, 53.06, 51.14, 30.78, 27.69, 21.15 ppm. HRMS: *m*/*z*: calcd for C_23_H_28_N_3_O_3_ [M + H]^+^: 394.2125, found: 394.2119. HPLC: t_R_ = 5.370 min (98 % at 254 nm).

##### 1-(6-Methoxy-1,5-naphthyridin-4-yl)-2-(4-((4-methylbenzyl)amino)cyclohexyl)ethan-1-ol (**8**)

The title compound was obtained according to general procedure using **D** (80 mg, 0.265 mmol, 1 eq), 4-methylbenzaldehyde (37 mL, 0.319 mmol, 1.2 eq), acetic acid, DCM (~5 mL) and Na(CH_3_COO)_3_BH (84 mg, 0.4398 mmol, 1.5 eq). The crude product was purified using flash column chromatography (SiO_2_, dichlorometane: methanol = 20: 1 to 9:1) to afford **8** (46 mg, 43%). ^1^H NMR (400 MHz, CDCl_3_): *δ* 8.65 (d, *J* = 4.5 Hz, 1H), 8.18 (d, *J* = 9.0 Hz, 1H), 7.40 (d, *J* = 4.5 Hz, 1H), 7.21 (d, *J* = 8.0, 2H), 7.13 (d, *J* = 7.8 Hz, 2H), 7.10 (d, *J* = 9.0 Hz, 1H), 4.07 (s, 3H), 3.79 (s, 2H), 3.40–3.27 (m, 2H), 3.03 (d, *J* = 11.9 Hz, 2H), 2.79–2.72 (m, 2H), 2.59–2.48 (m, 1H), 2.33 (s, 3H), 2.22–2.09 (m, 2H), 1.99–1.89 (m, 2H), 1.54–1.41 (m, 3H) ppm. ^13^C NMR (100 MHz, CDCl_3_): *δ* 161.40, 147.69, 146.83, 141.48, 141.00, 140.33, 137.70, 136.46, 129.11, 128.00, 124.23, 116.25, 58.48, 54.03, 53.70, 52.38, 50.50, 32.84, 28.49, 21.10 ppm. HRMS: *m*/*z*: calcd for C_25_H_32_N_3_O_2_ [M + H]^+^: 405.2416, found: 405.2410. HPLC: t_R_ = 4.977 min (100 % at 254 nm).

##### *N*-(4-Aminobenzyl)-1-(2-(6-methoxy-1,5-naphthyridin-4-yl)ethyl)piperidin-4-amine (**9**)

The title compound was obtained according to general procedure using **A** (250 mg, 0.873 mmol, 1 eq), 4-aminobenzaldehyde (127 mg, 1.048 mmol, 1.2 eq), acetic acid, DCM (~5 mL) and Na(CH_3_COO)_3_BH (278 mg, 1.309 mmol, 1.5 eq). The crude product was purified using flash column chromatography (SiO_2_, dichlorometane: methanol = 20: 1 to 9:1) to afford **9** (149 mg, 44%). ^1^H NMR (400 MHz, CDCl_3_): *δ* 8.64 (d, *J* = 4.5 Hz, 1H), 8.17 (d, *J* = 9.0 Hz, 1H), 7.39 (d, *J* = 4.5 Hz, 1H), 7.10 (dd, *J* = 8.7, 2.1 Hz, 3H), 6.67–6.61 (m, 2H), 4.06 (s, 3H), 3.70 (s, 2H), 3.62 (s, 1H), 3.39–3.30 (m, 2H), 3.02 (d, *J* = 11.9 Hz, 2H), 2.80–2.73 (m, 2H), 2.56–2.48 (m, 1H), 2.20–2.08 (m, 2H), 1.95–1.87 (m, 2H), 1.53–1.39 (m, 2H) ppm. ^13^C NMR (100 MHz, CDCl_3_): *δ* 13C 161.39, 147.68, 146.83, 145.26, 141.47, 140.99, 140.31, 130.73, 129.18, 124.23, 116.25, 115.15, 58.47, 53.97, 53.71, 52.40, 50.35, 32.81, 28.48 ppm. HRMS: *m*/*z*: calcd for C_23_H_30_N_5_O [M + H]^+^: 392.2372, found: 392.2438. HPLC: t_R_ = 3.723 min (94 % at 254 nm).

##### 4-(((4-(((6-methoxy-1,5-naphthyridin-4-yl)oxy)methyl)cyclohexyl)amino)methyl)aniline (**10**)

The title compound was obtained according to general procedure using **B** (110 mg, 0.383 mmol, 1 eq), 4-aminobenzaldehyde (56 mg, 0.459 mmol, 1.2 eq), acetic acid, DCM (~5 mL) and Na(CH_3_COO)_3_BH (122 mg, 0.574 mmol, 1.5 eq). The crude product was purified using flash column chromatography (SiO_2_, dichlorometane: methanol = 20: 1 to 9:1) to afford **10** (97 mg, 65%). ^1^H NMR (400 MHz, CDCl_3_): *δ* 8.59 (d, *J* = 5.2 Hz, 1H), 8.14 (d, *J* = 9.0 Hz, 1H), 7.10 (d, *J* = 8.9 Hz, 3H), 6.87 (d, *J* = 5.3 Hz, 2H), 6.67–6.62 (m, 2H), 4.09 (s, 3H), 4.01 (d, *J* = 6.2 Hz, 2H), 3.72 (s, 2H), 3.63 (s, 1H), 2.51 (s, 1H), 2.05 (d, *J* = 7.7 Hz, 5H), 1.29–1.12 (m, 4H) ppm. ^13^C NMR (100 MHz, CDCl_3_): *δ* 161.48, 160.04, 148.85, 145.26, 142.62, 139.99, 134.39, 130.80, 129.22, 116.48, 115.15, 104.97, 73.81, 56.29, 53.64, 50.76, 37.29, 32.78, 28.40 ppm. HRMS: *m*/*z*: calcd for C_23_H_29_N_4_O_2_ [M + H]^+^: 393.2285, found: 393.2278. HPLC: t_R_ = 4.693 min (100 % at 254 nm).

##### *N*-(4-Aminobenzyl)-6-(((6-methoxy-1,5-naphthyridin-4-yl)oxy)methyl)tetrahydro-2*H*-pyran-3-amine (**11**)

The title compound was obtained according to general procedure using **C** (160 mg, 0.553 mmol, 1 eq), 4-aminobenzaldehyde (80 mg, 0.664 mmol, 1.2 eq), acetic acid, DCM (~5 mL) and Na(CH_3_COO)_3_BH (176 mg, 0.829 mmol, 1.5 eq). The crude product was purified using flash column chromatography (SiO_2_, dichlorometane: methanol = 20: 1 to 9:1) to afford **11** (125 mg, 57%). ^1^H NMR (400 MHz, CDCl_3_): *δ* 8.59 (d, *J* = 5.2 Hz, 1H), 8.14 (d, *J* = 9.0 Hz, 1H), 7.11–7.06 (m, 3H), 6.91 (d, *J* = 5.3 Hz, 1H), 6.65–6.61 (m, 2H), 4.32–4.24 (m, 1H), 4.17–4.11 (m, 2H), 4.09 (s, 3H), 3.89–3.81 (m, 1H), 3.70 (s, 2H), 3.65 (s, 1H), 3.17 (t, *J* = 10.6 Hz, 1H), 2.79–2.70 (m, 1H), 2.17–2.10 (m, 1H), 2.00–1.91 (m, 1H) ppm. ^13^C NMR (100 MHz, CDCl_3_): *δ* 161.50, 159.76, 148.78, 145.47, 142.71, 140.00, 134.33, 130.35, 129.16, 116.58, 115.14, 105.45, 75.47, 72.70, 71.70, 53.70, 52.95, 50.97, 30.72, 27.64 ppm. HRMS: *m*/*z*: calcd for C_22_H_27_N_4_O_3_ [M + H]^+^: 395.2078, found: 395.20723. HPLC: t_R_ = 4.267 min (100 % at 254 nm).

##### *N*-(4-Bromo-3-(trifluoromethyl)benzyl)-1-(2-(6-methoxy-1,5-naphthyridin-4-yl)ethyl)piperidin-4-amine (**12**)

The title compound was obtained according to general procedure using **A** (220 mg, 0.768 mmol, 1 eq), 4-bromo-3-(trifluoromethyl)benzaldehyde (233 mg, 0.922 mmol, 1.2 eq), acetic acid, DCM (~5 mL) and Na(CH_3_COO)_3_BH (244 mg, 1.152 mmol, 1.5 eq). The crude product was purified using flash column chromatography (SiO_2_, dichlorometane: methanol = 20: 1 to 9:1) to afford **12** (189 mg, 47%). ^1^H NMR (400 MHz, CDCl_3_): *δ* 8.59 (d, *J* = 4.5 Hz, 1H), 8.12 (d, *J* = 9.0 Hz, 1H), 7.63 (d, *J* = 1.6 Hz, 1H), 7.57 (d, *J* = 8.2 Hz, 1H), 7.35 (d, *J* = 4.5 Hz, 1H), 7.32 (dd, *J* = 8.2, 1.5 Hz, 1H), 7.04 (d, *J* = 9.0 Hz, 1H), 4.01 (s, 3H), 3.77 (s, 2H), 3.35–3.29 (m, 2H), 3.00 (d, *J* = 11.7 Hz, 2H), 2.77–2.72 (m, 2H), 2.52–2.44 (m, 1H), 2.14 (t, *J* = 10.6 Hz, 2H), 1.90 (d, *J* = 11.4 Hz, 2H), 1.49–1.38 (m, 2H) ppm. “Amine H is exchangeable and is not visible on NMR spectrum.” ^13^C NMR (100 MHz, CDCl_3_): *δ* 161.41, 147.65, 146.57, 141.45, 140.95, 140.82, 140.28, 134.81, 132.48, 129.94 (q, *J* = 31.0 Hz), 127.35 (q, *J* = 5.3 Hz), 124.23, 121.65, 117.92, 116.29, 58.33, 54.27, 53.68, 52.22, 49.73, 32.68, 28.39 ppm. HRMS: *m*/*z*: calcd for C_24_H_27_BrF_3_N_4_O [M + H]^+^: 523.1315, found: 523.1297. HPLC: t_R_ = 5.013 min (98 % at 254 nm).

### 3.4. In Vitro DNA Gyrase and topoIV Inhibition

To determine the IC_50_ values of the compounds for *S. aureus* and *E. coli*, a DNA gyrase supercoiling high-throughput plate assay [29,30] and a topoIV relaxation high-throughput plate assay [31,32] were performed. Assay kits and black streptavidin-coated 96-well microtiter plates are available from Inspiralis (Norwich, UK). For pre-screening, an inhibitor concentration of 100 μM was used. For compounds showing residual activity less than 50% at 100 μM, IC_50_ values were calculated in GraphPad Prism 6.0 software. A nonlinear regression-based fit of the inhibition curves was used, with the log [inhibitor] versus slope of the reaction variables (four parameters) and a symmetric equation. Compounds that exhibited residual activity of more than 50% were labelled inactive. To determine IC_50_ values, two independent repetitions were performed and the average was calculated. The positive control was gepotidacin, which showed 0.374 µM and 0.244 µM for *S. aureus* and *E. coli* DNA gyrase, and 8.30 µM and 0.049 µM for *S. aureus* and *E. coli* TopoIV, as determined by our procedure. 

### 3.5. Human TopoIIα Selectivity Determination

For selectivity determination, a human topoisomerase II alpha relaxation high-throughput assay was used. The assay kits and black streptavidin-coated 96-well microtiter plates are available from Inspiralis (Norwich, UK) [33]. Two independent measurements were performed with an inhibitory concentration of 100 µM. Raw data were calculated as the mean ± SD percentage of the residual enzyme activity.

### 3.6. Antimicrobial Activity Assay

For the determination of minimum inhibitory concentrations (MIC), the broth microdilution method was used in 96-well plate format, according to the guidelines of the Clinical and Laboratory Standards Institute [34] and the recommendations of the European Committee on Antimicrobial Susceptibility Testing (EUCAST) [35]. The dilutions of the bacterial suspension of the specific bacterial strain, corresponding to the 0.5 McFarland turbidity standard, were prepared with cation-matched Mueller–Hinton broth with TES (Thermo Fisher Scientific) to obtain an end inoculum of 10^5^ CFU/mL. The compounds were dissolved in DMSO and mixed with the inoculum. The mixture was incubated at 35 °C for 20 h, and MIC values were determined visually. The MIC were the lowest dilutions of compounds that did not exhibit turbidity. The MIC was determined as the lowest concentration of antibiotic that prevented visible bacterial growth. Tetracycline was used as a positive control on each test plate.

### 3.7. Metabolic Activity Assessment

HUVEC (ATCC^®^ CRL-1730™) and HepG2 (ATCC^®^ HB-8065™) cell lines were cultured in Dulbecco’s Modified Eagle Medium (DMEM; Sigma-Aldrich, St. Louis, MO, USA) supplemented with 10% heat-deactivated FBS (Gibco, Grand Island, NE, USA), 2 mM L-glutamine, 100 U/mL penicillin, 100 μg/mL streptomycin (all from Sigma-Aldrich, St. Louis, MO, USA) in a humidified chamber at 37 °C and 5% CO_2_. The experiments were carried out on passages 4–7 for both cell lines. The tested compounds were dissolved in DMSO and further diluted in culture medium to a desired final concentration. The cells were then seeded into 96-well plates at 5 × 10^4^ cells/mL (100 μL/well) or 8 × 10^4^ cells/mL (100 μL/well) for HUVEC and HepG2, respectively. The cells were then incubated for 24 h to allow attachment onto the wells. The cells were treated with 1 μM and 50 μM of each compound of interest, or the corresponding vehicle as control. For IC_50_ determination, the cells were seeded into 96-well plates, and after attachment, treated with nine different concentrations (the final concentrations were in the range of 0.39–100 μM) of the selected compounds, or the corresponding vehicle as control. Their metabolic activity was assessed after 72 h treatment using a CellTiter 96 Aqueous One Solution Cell Proliferation Assay (Promega, Madison, WI, USA), in accordance with the manufacturer’s instructions. The absorbance was measured at 492 nm on an automated microplate reader, Tecan Spark (Tecan Group Ltd., Männedorf, CH, Switzerland). Data are presented as the percentage of metabolic activity of control cells stimulated with vehicle, or as an IC_50_ value (mean ± SD) from three independent experiments, each conducted in triplicate (for screening) or duplicate (for IC_50_ determination). IC_50_ values were calculated using a non-linear regression model and with Graph Pad Prism 9 software.

## 4. Conclusions

In summary, with the newly designed series of NBTIs presented in this study, we have probed whether alternative interactions may offer improvements to the well-established halogen bond-forming moieties exemplified in our previous work. Molecular docking calculations predicted that the geometry of the amino group of the *p*-amino phenyl RHS moiety (in particular, the interatomic distances and angles) is simply not suitable for establishing hydrogen-bonding interactions with the backbone carbonyl oxygens of the *S. aureus* GyrA A68 residues. In vitro biological evaluation of the NBTIs containing *p*-amino phenyl RHS showed no measurable antibacterial activity (MIC > 128 µg/mL) and enzyme inhibition potency (IC_50_ > 100 µM) in both Gram-positive and Gram-negative bacteria, and confirmed the prediction of the molecular docking calculations. In contrast, hydrophobic interactions can occur mainly in Gram-positive topoisomerases, and are apparently favourable for NBTI’s antibacterial activity against *A. baumannii*. This clearly pinpoints that van der Waals/hydrophobic interactions offer a valid alternative to the halogen bonds at NBTI’s binding site. However, these interactions do not match the halogen-bonding interactions in terms of the NBTI’s enzyme inhibitory potency, and consequently, they induce less potent antibacterial activity, as demonstrated in our previously published series of halogenated NBTI analogues. 

## Data Availability

The data presented in this study are available on request from the corresponding authors.

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
