# Peer review of "Exploring Alternative Pathways to Target Bacterial Type II Topoisomerases Using NBTI Antibacterials: Beyond Halogen-Bonding Interactions"

_antibiotics, 2023, doi:10.3390/antibiotics12050930_

Round 1

Reviewer 1 Report

The manuscript from Kokot et al. describes the in silico and in vitro evaluation of a series of some novel bacterial topoisomerase inhibitors. The showed results fitted the journal aims and are well described. Some of them display a good antibacterial activity and the authors indicate the DNA gyrase as target of the compounds. A major concern is the lack of any data regarding the safety of these compounds, thus at least one eukaryotic cell line must be assayed in order to exclude a lethal activity that could impede further applications of the compounds. Moreover, since the authors chose a fluorescence-based Gyrase/Topoisomerase assay for IC50 determination, instead of the gel-based assay, it would be better to perform the latter assay, at least for the most active compound(s), in order to visualize the inhibitory activity toward the gyrase activity on a plasmid DNA, at the calculated IC50 value. Finally, the authors should explain the reason(s) why compounds 5,6 and 7 (which differ only for the linker, not for RHS) possess a similar IC50 in inhibiting the E.coli gyrase but only compound 5 has a good MIC value against E.coli, whereas 6 and 7 have no activity (MIC >128). The opposite happens for S.aureus, where all the compound are good gyrase inhibitors and have similar MIC values. As well compound 9, which bears an -NH2 substituent on RHS and has a different linker, possesses a good IC50 value in inhibiting S.aureus gyrase and no activity against that of E.coli,  and then high MIC values (or ND) against the relative bacteria. This could seem paradoxical and deserves a further evaluation/explanation.

Minor

Please, add the meaning of asterisks, superscripts and abbreviations in table 2, as done for table 1.

Please, check for typos

Author Response

The manuscript from Kokot et al. describes the in silico and in vitro evaluation of a series of some novel bacterial topoisomerase inhibitors. The showed results fitted the journal aims and are well described. Some of them display a good antibacterial activity and the authors indicate the DNA gyrase as target of the compounds.

We would like to thank the referee for the critical review of the manuscript and all the suggestions.

Remark 1:

A major concern is the lack of any data regarding the safety of these compounds, thus at least one eukaryotic cell line must be assayed in order to exclude a lethal activity that could impede further applications of the compounds.

Reply 1:

To obtain data on the safety of the compounds, we performed analyses on HepG2 na HUVEC cells to test the cytotoxicity of the compounds. The results are shown in Table 3 (line 204).

Remark 2:

Moreover, since the authors chose a fluorescence-based Gyrase/Topoisomerase assay for IC50 determination, instead of the gel-based assay, it would be better to perform the latter assay, at least for the most active compound(s), in order to visualize the inhibitory activity toward the gyrase activity on a plasmid DNA, at the calculated IC50 value.

Reply 2:

For all our NBTIs, we determine IC50 values using a fluorescence-based Gyrase/Topoisomerase assay, Therefore we think it is most representative to also test this set of compounds with the same assay to ensure comparability of results for the new set of compounds to the previous compounds.

 Remark 3:

Finally, the authors should explain the reason(s) why compounds 5,6 and 7 (which differ only for the linker, not for RHS) possess a similar IC50 in inhibiting the E.coli gyrase but only compound 5 has a good MIC value against E.coli, whereas 6 and 7 have no activity (MIC >128). The opposite happens for S.aureus, where all the compound are good gyrase inhibitors and have similar MIC values.

Reply 3:

Thank you for pointing this out, we have explained the possible reasons in the manuscript in line 163:

Although compounds 5, 6, and 7 show similar inhibition of E. coli DNA gyrase, only compound 5 has potent antibacterial activity against E. coli. Compounds with cyclohexane (6) and tetrahydropyran (7) linkers may have lower permeability and are more easily pumped-out by the efflux pumps. Compared with wild-type E. coli, the activity was enhanced in the AcrA knock-out E. coli N43 strain and in E. coli D22 strain with a mutation in the lpxC gene that increases membrane permeability. This suggests that the membrane permeability of the compounds is suboptimal for crossing the membrane of Gram-negative pathogens and, more importantly, that they are efficiently pumped- out by the E. coli efflux pumps.

 Remark 4:

As well compound 9, which bears an -NH2 substituent on RHS and has a different linker, possesses a good IC50 value in inhibiting S. aureus gyrase and no activity against that of E.coli,  and then high MIC values (or ND) against the relative bacteria. This could seem paradoxical and deserves a further evaluation/explanation.

 Reply 4:

Antibacterial activity (MIC values) depends on inhibition of both enzymes (IC50s) as well as on membrane permeability in the bacteria and efflux of the compounds with bacteria efflux pumps. Due to numerous possible factors, it is not possible to specify only one reason.

 Remark 5:

Please, add the meaning of asterisks, superscripts and abbreviations in table 2, as done for table 1.

Please, check for typos.

Reply 5:

Thank you for these observations, we added the footnote under the Table 2.

Reviewer 2 Report

1] Its better to include the structures of the synthesized compounds supporting information. It can be included below the name of the structure along with the spectral and analytical data representing the compound.

2] Synthetic scheme is missing. It has to be included under Materials and Methods.

3] A summary of SAR needs to be included at the end of Results and Discussion. 

4] Authors are encouraged to include the Physichochemical properties of the most potent compounds and discuss the data in correlation to the biological activity. They can use the SWISSADME website if required.

5] Mass spectral data is reported in the manuscript, however the spectras are missing in Supporting Information. It is suggested to include them in the document.

6] Under Introduction, its better to include some literature compounds in this area.

The manuscript can be accepted after said corrections are implemented.

Author Response

We would like to thank the referee for the critical review of the manuscript and all the suggestions.

Remark 1:

Its better to include the structures of the synthesized compounds supporting information. It can be included below the name of the structure along with the spectral and analytical data representing the compound.

Reply 1:

Thank you for this suggestion. However, we think it is clearer if the structures of the compounds are in the manuscript.

Remark 2:

Synthetic scheme is missing. It has to be included under Materials and Methods.

Reply 2:

We added the synthetic scheme in the Materials and Method section (line 255).

 Remark 3:

A summary of SAR needs to be included at the end of Results and Discussion. 

Reply3:

We have added SAR of the new NBTIs at the end of Results and Discussion (line 218). We have also added Figure 4 to better explain it.

“Different RHS moieties form different interactions with the amino acid residues of the binding pocket, resulting in a different structure-activity relationship (SAR). As shown in Figure 4, NBTIs with p-bromophenyl RHSs show inhibition for three out of four enzymes due to hydrophobic interactions and additional strong halogen bonds in the binding pocket. Compounds with p-methylphenyl RHSs that form a hydrophobic interaction show inhibition, but it is weaker. Whereas p-aminophenyl RHS NBTIs do not form interaction with amino acid residues in the binding pocket, therefore no inhibition was observed.”

 Remark 4:

Authors are encouraged to include the Physichochemical properties of the most potent compounds and discuss the data in correlation to the biological activity. They can use the SWISSADME website if required.

Reply 4:

We have included the table (Table 4, line 215) with the physicochemical properties and also discuss the obtained data in correlation with the biological activity (line 208).

“The SWISSADME tool [27] was used to calculate the physicochemical properties of the newly synthesized NBTIs (Table 3). With the exception of compound 12, they exhibit similar physicochemical properties. Despite the higher molecular weight, larger number of heavy atoms, rotatable bonds, and H-bond acceptors, it shows strong biological activity. The major difference between compounds with p-methyl phenyl RHSs (58) and p-amino phenyl RHSs (911) is in their topological polar surface area (TPSA) parameter, suggesting that a larger polar surface area leads to lower biological activity.”

 Remark 5:

Mass spectral data is reported in the manuscript, however the spectra are missing in Supporting Information. It is suggested to include them in the document.

Reply 5:

We added the HRMS spectra of the compounds in Supporting Information.

 Remark 6:

Under Introduction, its better to include some literature compounds in this area.

Reply 6:

We added figure with representatives of NBTIs (line 64).

Reviewer 3 Report

Line 56: ParC is a subunit of Topo IV

You only looked at changes in catalytic activity of the enzyme (relaxation/supercoiling). Buildup of DNA breaks is primarily what leads to cell death. Would suggest adding cleavage data to the manuscript. NBTIs primarily lead to single-stranded DNA breaks. 

The manuscript refers back to the previous work and it makes the conclusions hard to make that it is truly the hydrogen bonding interactions that is leading to the changes in activity and interactions with the enzyme. I would try to make it more clear in this manuscript what data you are using to make these conclusions.

Author Response

We would like to thank the referee for the critical review of the manuscript and all the suggestions.

Remark 1:

Line 56: ParC is a subunit of Topo IV

Reply 1:

Thank you for this observation. We noticed now that it was not clearly written. We corrected the sentence from (highlighted in grey):

“non-catalytic hydrophobic binding pocket formed at the interface of both GyrA, i.e., ParC subunits of DNA  gyrase, i.e., topoIV enzyme, respectively, and a central linker moiety connecting both parts [13,14].” to

“non-catalytic hydrophobic binding pocket formed at the interface of both GyrA subunits of DNA gyrase, or both ParC subunits of topoIV enzyme, and a central linker moiety connecting both parts [13,14].”

Remark 2:

You only looked at changes in catalytic activity of the enzyme (relaxation/supercoiling). Buildup of DNA breaks is primarily what leads to cell death. Would suggest adding cleavage data to the manuscript. NBTIs primarily lead to single-stranded DNA breaks. 

Reply 2:

In our previous publication Kolarič et al [22], we performed a cleavage experiment. Since these new NBTIs are structurally similar to the NBTIs published in Kolarič et al [22],  we assume that they stabilize single-strand DNA breaks in a similar way as the co-crystallized AMK-12 ligand in the DNA gyrase of S. aureus in Kolarič et al [22]. We also explained this in the manuscript (line 184):

“Since, NBTIs stop the enzymatic reaction at the level of single-strand DNA breaks, there is a correlation between IC50 supercoiling and single-strand breaks. They disrupt the spatial symmetry and topology of nascent DNA due to the asymmetry of the 6-methoxy-1,5-naphthyridine LHS intercalated between DNA base pairs. Consequently they stabilize single-strand DNA breaks in DNA gyrase in a similar manner as the majority of NBTIs, as we confirmed for structurally similar NBTIs in a DNA gyrase cleavage assay in our previous publication [22].”

Remark 3:

The manuscript refers back to the previous work and it makes the conclusions hard to make that it is truly the hydrogen bonding interactions that is leading to the changes in activity and interactions with the enzyme. I would try to make it more clear in this manuscript what data you are using to make these conclusions.

Reply 3:

In this manuscript, we have explained that the newly designed NBTI analogues are NOT capable of any hydrogen bonding with amino acid residues in the NBTI binding site. We have explained this in the discussion and also mentioned it in the conclusion (highlighted in grey):

Molecular docking calculations predicted that the geometry of the amino group of the p-amino phenyl RHS moiety and in particular the interatomic distances and angles are simply not suitable for establishing hydrogen-bonding interactions with the backbone carbonyl oxygens of the S. aureus GyrA A68 residues.

Round 2

Reviewer 1 Report

Authors did not perform all the requested experiments. Safety studies and explanations about the determined antibacterial activity are not convincing.

Author Response

Dear reviewer,

We performed all the reasonable experiments suggested by the reviewer. We performed cytotoxicity studies not only on one but on two eukaryotic cell lines. We added a paragraph describing the results of cytotoxicity:

According to cytotoxicity assessments on the human umbilical vein endothelial cells (HUVECs) and HepG2 liver cancer cell lines (Table 3), compounds 5, 8, and 12 showed weaker effects by one to three orders of magnitude on human cells compared to antibacterial activity against the Gram-positive bacteria (Table S3) and the same order of magnitude compared to the Gram-negative bacteria (Table S3). Due to weak antibacterial activity of other compounds (6, 7, 911) against Gram-positive and Gram-negative bacteria, MICs were approximately of the same order of magnitude as their cytotoxic IC50 values.

Furthermore, Reviewer 1 requested to perform gel-based assay instead of a fluorescence-based Gyrase-Topoisomerase assay for IC50 determination. Although it is true that gel-based assay provides the inhibitory activity on a plasmid DNA, fluorescence based-assay is used for screening purposes by many authors and since IC50 values are relative values that should be used for comparison with standard compounds (like gepotidacin or other inhibitors with determined Ki, which we did), and fluorescence-based assay allows this relative comparison. Taken this into account, we believe that results obtained with gel-based assay would not offer significant additional value, especially since compounds from this library show substantial toxicity on human cell lines and will enter the hit-to-lead phase.

Reviewer 3 Report

The edits are sufficient and recommend for publication.

Author Response

We thank the reviewer for his suggestions and positive feedback on our manuscript.